# WHY LOTTERY TICKET WINS?
# A THEORETICAL PERSPECTIVE OF SAMPLE COMPLEXITY ON SPARSE NEURAL NETWORKS

## ABSTRACT

The *lottery ticket hypothesis* (LTH) (Frankle & Carbin, 2019) states that learning on a properly pruned network (the *winning ticket*) has improved test accuracy over the originally unpruned network. Although LTH has been justified empirically in a broad range of deep neural network (DNN) involved applications like computer vision and natural language processing, the theoretical validation of the improved generalization of a winning ticket remains elusive. To the best of our knowledge, our work, for the first time, characterizes the performance of training a sparse neural network by analyzing the geometric structure of the objective function and the sample complexity to achieve zero generalization error. We show that the convex region near a desirable model with guaranteed generalization enlarges as the neural network model is pruned, indicating the structural importance of a winning ticket. Moreover, as the algorithm for training a sparse neural network is specified as (accelerated) stochastic gradient descent algorithm, we theoretically show that the number of samples required for achieving zero generalization error is proportional to the number of the non-pruned weights in the hidden layer. With a fixed number of samples, training a pruned neural network enjoys a faster convergence rate to the desirable model than training the original unpruned one, providing a formal justification of the improved generalization of the winning ticket. Our theoretical results are acquired from learning a sparse neural network of one hidden layer, while experimental results are further provided to justify the implications in pruning multi-layer neural networks.

## 1 INTRODUCTION

Neural network pruning can reduce the computational cost of training a model significantly (LeCun et al., 1990; Hassibi & Stork, 1993; Dong et al., 2017; Han et al., 2015; Hu et al., 2016; Srinivas & Babu, 2015; Yang et al., 2017; Molchanov et al., 2017). The recent *Lottery Ticket Hypothesis* (LTH) (Frankle & Carbin, 2019) claims that a randomly initialized dense neural network always contains a so-called "winning ticket," which is a sub-network bundled with the corresponding initialization, such that when trained in isolation, this winning ticket can achieve at least the same testing accuracy as that of the original network by running at most the same amount of training time. This so-called "improved generalization of winning tickets" is verified empirically in (Frankle & Carbin, 2019). LTH has attracted a significant amount of recent research interests (Ramanujan et al., 2020; Zhou et al., 2019; Malach et al., 2020). Despite the empirical success (Evci et al., 2020; You et al., 2019; Wang et al., 2019; Chen et al., 2020a), the theoretical justification of winning tickets remains elusive except for a few recent works. Malach et al. (2020) provides the first theoretical evidence that within a randomly initialized neural network, there exists a good sub-network that can achieve the same test performance as the original network. Meanwhile, recent work (Neyshabur, 2020) trains neural network by adding the $\ell_1$ regularization term to obtain a relatively sparse neural network, which

has a better performance numerically. Arora et al. (2018) and Zhou et al. (2018) show that the expressive power of a neural network is comparable to a compressed neural network and both networks have the same generalization error. However, no theoretical explanation has been provided for the improved generalization of winning tickets, i.e., winning tickets can achieve higher test accuracy after the same training time.

**Contributions**: This paper provides the *first* systematic analysis of learning sparse neural networks with a finite number of training samples. Our analytical results also provide justification of the LTH from the perspective of the sample complexity. In particular, we provide the *first* theoretical justification of the improved generalization of winning tickets. Specific contributions include:

1. **Sparse neural network learning via accelerated gradient descent (AGD)**: We propose an AGD algorithm with tensor initialization to learn the sparse model from training samples. Considering the scenario where there exists a ground-truth sparse one-hidden-layer neural network, we prove that our algorithm converges to the ground-truth model linearly, which has guaranteed generalization on testing data.

2. **First sample complexity analysis for pruned networks**: We characterize the required number of samples for successful convergence, termed as the *sample complexity*. Our sample complexity depends linearly on the number of the non-pruned weights of the sparse network and is a significant reduction from directly applying conventional complexity bounds in (Zhong et al., 2017; Zhang et al., 2020a;c).

3. **Characterization of the benign optimization landscape of pruned networks**: We show analytically that the empirical risk function has an enlarged convex region near the ground-truth model if the neural network is sparse, justifying the importance of a good sub-network (i.e., the winning ticket).

4. **Characterization of the improved generalization of winning tickets**: We show that gradient-descent methods converge faster to the ground-truth model when the neural network is properly pruned, or equivalently, learning on a pruned network returns a model closer to the ground-truth model with the same number of iterations, indicating the improved generalization of winning tickets.

## 1.1 RELATED WORK

**Winning tickets**. Frankle & Carbin (2019) proposes an *Iterative Magnitude Pruning* (IMP) algorithm to obtain the proper sub-network and initialization. IMP and its variations (Frankle et al., 2019a; Renda et al., 2019) succeed in deeper networks like Residual Networks (Resnet)-50 and Bidirectional Encoder Representations from Transformers (BERT) network (Chen et al., 2020b). (Frankle et al., 2019b) shows that IMP succeeds in finding the "winning ticket" if the ticket is stable to stochastic gradient descent noise. In parallel, (Liu et al., 2018) shows numerically that the "winning ticket" initialization does not improve over a random initialization once the correct subnetworks are found, suggesting that the benefit of "winning ticket" mainly comes from the sub-network structures.

**Over-parameterized model.** When the number of weights in a neural network is much larger than the number of training samples, the landscape of the objective function of the learning problem has no spurious local minima, and first-order algorithms converge to one of the global optima (Livni et al., 2014; Zhang et al., 2016; Soltanolkotabi et al., 2018). However, the global optima is not guaranteed to generalize well on testing data (Yehudai & Shamir, 2019; Zhang et al., 2016).

**Model estimation & Generalization analysis.** This framework assumes a ground-truth model that maps the input data to the output labels, and the learning objective is to estimate the ground-truth model, which has a generalization guarantee on testing data. The learning problem has intractably many spurious local minina even for one-hidden-layer neural networks (Shamir, 2018; Safran & Shamir, 2018; Zhang et al., 2016). Assuming an infinite number of training samples, (Brutzkus & Globerson, 2017; Du et al., 2018; Tian, 2017) develop learning methods to estimate the ground-truth model. (Fu et al., 2018; Zhong et al., 2017; Zhang et al., 2020a;c) extend to the practical case of a finite number of samples and characterize the

sample complexity for successful estimation of the ground-truth model. Because the analysis complexity explodes when the number of hidden layers increases, all the analytical results about estimating the ground-truth model are limited to one-hidden-layer neural networks, and the input distribution is often assumed to be the standard Gaussian distribution.

## 1.2 NOTATIONS

Vectors are bold lowercase, matrices and tensors are bold uppercase. Scalars are in normal font, and sets are in calligraphy and blackboard bold font. $I$ and $e_i$ denote the identity matrix and the $i$-th standard basis vector. $[Z]$ stands for the set of $\{1, 2, \cdots, Z\}$ for any number $\mathbb{N}^+$. In addition, $f(r) = \mathcal{O}(g(r))$ (or $f(r) = \Omega(g(r))$) if $f \leq C \cdot g$ (or $f \geq C \cdot g$) for some constant $C > 0$ when $r$ is large enough. $f(r) = \Theta(g(r))$ if both $f(r) = \mathcal{O}(g(r))$ and $f(r) = \Omega(g(r))$ holds, where $c \cdot g \leq f \leq C \cdot g$ for some constant $0 \leq c \leq C$ when $r$ is large enough.

## 2 PROBLEM FORMULATION

Given $N$ pairs of training samples $\mathcal{D} = \{x_n, y_n\}_{n=1}^N$ where $x_n \in \mathbb{R}^d$ represent input features and $y_n \in \mathbb{R}$ represents the corresponding output label. We consider a teacher-student setup where for any input $x$, the corresponding output $y$ is generated by a sparse one-hidden-layer neural network, which is called *teacher*, as shown in Fig. 1. Specifically, the teacher network is equipped with $K$ neurons where each neuron is connected to $r^*$ $(r^* \leq d)$[1] input features. Let $\widehat{W}^* \in \mathbb{R}^{r \times K}$ contain all the ground truth weights of the $K$ neurons in the teacher network, where the $j$-th column, denoted by $\widehat{w}_j^* \in \mathbb{R}^{r^*}$, contains the edge weights connected to the $j$-th neuron. Let $\Omega_j^*$ denote the set of indices of the input features to which the $j$-th neuron is connected in the teacher network.

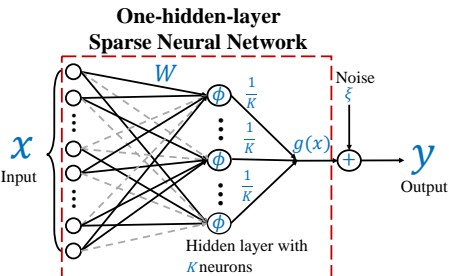

**One-hidden-layer Sparse Neural Network**

Figure 1: Illustration of the model

Given input $x_n$, let $x_{n,\Omega_j^*} \in \mathbb{R}^r$ represent the sub-vector that is connected to $j$-th neuron, where $j \in [K]$. Then, in a regression problem, $y_n$ is obtained by the teacher network through [2]

$$y_n = \frac{1}{K} \sum_{j=1}^K \phi(\widehat{w}_j^{*T} x_{n,\Omega_j^*}) + \xi_n := g(x_n; \widehat{W}^*) + \xi_n, \tag{1}$$

where $\xi_n$ is the unknown additive noise and $\phi$ is the activation function.

We train on a *student* network equipped with same $K$ neurons as in the teacher network, however, each neuron in the student network is connected to $r$ input features instead of $r^*$. Let $\Omega_j$ denote the set of indices of the input features to which the $j$-th neuron is connected in the student network. Then, given any estimated weights $W = [w_1, w_2, \cdots, w_K] \in \mathbb{R}^{r \times K}$ for the student networks, the empirical risk function with respect to the training set $\mathcal{D}$ is defined as

$$\hat{f}_{\mathcal{D}}(W) = \frac{1}{2N} \sum_{n=1}^N \left( \frac{1}{K} \sum_{j=1}^K \phi(w_j^T x_{n,\Omega_j}) - y_n \right)^2, \tag{2}$$

---

[1]We consider the same $r^*$ for each neuron in this paper, but the analysis can be easily modified to handle the case of different numbers of features per neuron.

[2]The analysis is extendable to binary classification, and the output is generated by $\text{Prob}(y_n = 1 | x_n) = g(x_n; \widehat{W}^*)$.

where we consider the square loss function here. We focus on the case that $r \geq r^*$ and $\Omega_j \supseteq \Omega_j^*$. We will discuss how to obtain such a student network from a fully connected neural network in Section 4.1.

Hence, the learning objective is to estimate a proper weight matrix $\boldsymbol{W}$ for the student network from the training samples $\mathcal{D}$ via solving the following problem:

$$\min_{\boldsymbol{W} \in \mathbb{R}^{r \times K}} \quad \hat{f}_{\mathcal{D}}(\boldsymbol{W}). \tag{3}$$

Let us define an augmented matrix $\boldsymbol{W}^* \in \mathbb{R}^{r \times K}$ such that $\boldsymbol{W}_{i,j}^* = \widehat{\boldsymbol{W}}_{i,j}^*$ if $i$ belongs to both $\Omega_j$ and $\Omega_j^*$ and $0$ otherwise. Clearly, $\boldsymbol{W}^*$ is a global minimizer to both (3) and its expectation $\mathbb{E}_{\mathcal{D}} \hat{f}_{\mathcal{D}}(\boldsymbol{W})$.

when measurements are noiseless, i.e., $\xi_n = 0$ for all $n$. Hence, if one can estimate $\boldsymbol{W}^*$ accurately from the training data on the student network, one can find the teacher network with $\widehat{\boldsymbol{W}}^*$ equivalently, which has guaranteed generalization performance on the testing data.

Following the standard assumption in (Zhong et al., 2017), $\boldsymbol{x}_n$ is independent and identically distributed (i.i.d.) from the standard Gaussian distribution $\mathcal{N}(\boldsymbol{0}, \boldsymbol{I}_{d \times d})$. [3] Throughout the paper, we consider rectified linear unit (ReLU), where $\phi(z) = \max\{z, 0\}$. $\xi_n$ can be arbitrary, and we only assume $\xi_n$ is bounded as $|\xi_n| \leq |\xi|$ for some constant $\xi$.

The questions that this paper addresses include: (1) **what algorithm** to estimate the augmented ground truth weights $\boldsymbol{W}^*$? (2) what is the **sample complexity** for the accurate estimate? (3) what is the **impact of the network pruning** on the difficulty of the learning problem and the performance of the learned model?

**Connections with conventional pruned networks.** To connect the student network with conventional pruned networks, we define a further augmented matrix $\widetilde{\boldsymbol{W}}^* \in \mathbb{R}^{d \times K}$, where the $j$-th column of $\widetilde{\boldsymbol{W}}^*$, denoted by $\widetilde{\boldsymbol{w}}_j^*$, contains the weights connected to the $j$-th neuron, and $\widetilde{\boldsymbol{W}}^*$ is zero padded in other entries for non-existing weights. In other words, $\widetilde{\boldsymbol{w}}_{j,\Omega_j}^* = \boldsymbol{w}_j^*$ for all $j \in [K]$, and $\widetilde{W}_{ij}^* = 0$ if $i \notin \Omega_j$. The mask matrix $\boldsymbol{M}$, as discussed in (Frankle & Carbin, 2019), with respect to $\boldsymbol{W}^*$ is defined as $M_{ij} = 1$ if $i \in \Omega_j$, and $M_{ij} = 0$ otherwise. Then, we have $\boldsymbol{M} \odot \widetilde{\boldsymbol{W}}^* = \widetilde{\boldsymbol{W}}^*$, where $\odot$ stands for the entry-wise multiplication. In (Frankle & Carbin, 2019), an Iterative Magnitude Pruning (IMP) algorithm is proposed to compute the mask matrix $\boldsymbol{M}$ for the lottery ticket. Equation (3) can be equivalently written as

$$\min_{\widetilde{\boldsymbol{W}} \in \mathbb{R}^{d \times K}} \hat{f}_{\mathcal{D}}(\widetilde{\boldsymbol{W}}) = \frac{1}{2N} \sum_{n=1}^{N} \left( \frac{1}{K} \sum_{j=1}^{K} \phi(\widetilde{\boldsymbol{w}}_j^T \boldsymbol{x}_n) - y_n \right)^2, \qquad \text{s.t. } \widetilde{\boldsymbol{W}} = \boldsymbol{M} \odot \widetilde{\boldsymbol{W}}, \tag{4}$$

where $\widetilde{\boldsymbol{W}} = [\widetilde{\boldsymbol{w}}_1, \widetilde{\boldsymbol{w}}_2, \cdots, \widetilde{\boldsymbol{w}}_K] \in \mathbb{R}^{d \times K}$ is an estimate of the augmented weight matrix $\widetilde{\boldsymbol{W}}^*$. $\widetilde{\boldsymbol{W}}^*$ is a global minimizer to (4) when measurements are noiseless. Note that (4) differs from the learning problem of the conventional dense networks in the additional constraint $\boldsymbol{M} \odot \widetilde{\boldsymbol{W}} = \widetilde{\boldsymbol{W}}$. Back to the problem setup in this paper, suppose $\boldsymbol{M}^*$ denotes the mask of the teacher model, then we learn on a student network with mask $\boldsymbol{M}$ satisfying $\text{Supp}(\boldsymbol{M}) \supseteq \text{Supp}(\boldsymbol{M}^*)$, where $\text{Supp}(\cdot)$ stands for the indices of the non-zero entries. Moreover, when $\boldsymbol{M}$ is an all-one matrix, i.e., $r = d$, (4) reduces to the conventional learning problem of a one-hidden-layer neural network.

We will mainly focus on the case that a proper student network is given with $r \geq r^*$ and $\Omega_j \supseteq \Omega_j^*$ and will discuss how to obtain a proper student network by the IMP algorithm in Section 4.1.

---

[3]The assumption is critical in theoretical analysis since the proof is built upon bounding the population risk function, and the Gaussian assumption is leveraged to analyze the landscape of the population risk function. It is possible to extend the analysis to other distributions and we will leave that for future work.

## 3 LOCAL GEOMETRIC STRUCTURE

We defer the algorithmic design and analysis to Section 4 and study the geometric structure of (2) here. Theorem 1 characterizes the local convexity of $\hat{f}_{\mathcal{D}}(\boldsymbol{W})$ in (2). It has two important implications.

1. **Strictly local convex ball around $\boldsymbol{W}^*$**: $\hat{f}_{\mathcal{D}}(\boldsymbol{W})$ is strictly convex around $\boldsymbol{W}^*$, and the radius of the convex ball is negatively correlated with $\sqrt{r}$, where $r$ is the number of non-pruned weights per neuron. Thus, the convex ball enlarges as $r$ decreases.

2. **Importance of the winning ticket architecture**: Compared with training on the dense network directly, training on a correctly pruned sub-network has a larger convex region near $\boldsymbol{W}^*$, which may lead to easier estimation of $\boldsymbol{W}^*$. To some extend, this result can be viewed as a theoretical validation of the importance of the winning architecture (a good sub-network) in (Frankle & Carbin, 2019). Formally, we have

**Theorem 1** (Local Convexity). *Suppose constants $\varepsilon_0$, $\varepsilon_1 \in (0, 1)$. When the number of samples satisfies*

$$N = \Omega\big(\varepsilon_1^{-2} K^4 r \log d\big), \tag{5}$$

*then for any $\boldsymbol{W}$ that satisfies*

$$\|\boldsymbol{W} - \boldsymbol{W}^*\|_F = \mathcal{O}\left(\frac{\varepsilon_0}{K^2}\right), \tag{6}$$

*we have*

$$\Theta\left(\frac{1 - \varepsilon_0 - \varepsilon_1}{K^2}\right)\boldsymbol{I} \preceq \nabla^2 \hat{f}_{\mathcal{D}}(\boldsymbol{W}) \preceq \Theta\left(\frac{1}{K}\right)\boldsymbol{I}. \tag{7}$$

Theorem 1 shows that with enough samples as shown in (5), in a local region of $\boldsymbol{W}^*$ as shown in (6), all the eigenvalues of Hessian matrix of the empirical risk function are lower and upper bounded by two positive constants. This property is useful in designing efficient algorithms to recover $\boldsymbol{W}^*$, as shown in Section 4.

Moreover, when the number of samples $N$ is fixed and $r$ changes, $\varepsilon_1$ can be $\Theta(\sqrt{r/N})$ while (5) is still met. $\varepsilon_0$ in (7) can be arbitrarily close to but small than $1 - \varepsilon_1$ so that the Hessian matrix is still positive definite. Then from (6), the radius of the convex ball is $\Theta(1) - \Theta(\sqrt{r/N})$, indicating an enlarged region when $r$ decreases.

## 4 CONVERGENCE ANALYSIS WITH ACCELERATED GRADIENT DESCENT

We propose to solve the non-convex problem (3) via the accelerated gradient descent (AGD) algorithm, summarized in Algorithm 1. Compared with the vanilla gradient descent (GD) algorithm, AGD has an additional momentum term, denoted by $\beta(\boldsymbol{W}^{(t)} - \boldsymbol{W}^{(t-1)})$, in each iteration. AGD enjoys a faster convergence rate than GD in solving optimization problems including learning neural networks (Zhang et al., 2020b). Vanilla GD can be viewed as a special case of AGD by letting $\beta = 0$.

The initial point $\boldsymbol{W}^{(0)}$ can be obtained through a tensor initialization method, which is built upon Algorithm 1 in (Zhong et al., 2017) for fully connected neural networks with modification to handle a sparse neural network. Specifically, we reduce the complexity dependence from input data dimension $d$ to the sparsity $r$, the definitions of the high-order moments (see (13)-(15) in Appendix A) are modified by replacing $\boldsymbol{x}$ in Definition 5.1 in Zhong et al. (2017) with $\tilde{\boldsymbol{x}} = \frac{1}{\sqrt{K}}\sum_{j=1}^{K}\boldsymbol{x}_{\Omega_j} \in \mathbb{R}^r$. Details of the tensor initialization method are provided in Appendix A.

The theoretical analyses of our algorithm are summarized in Theorem 2 (convergence) and Lemma 1 (Initialization). The significance of these results can be interpreted from the following aspects.

---

**Algorithm 1** Accelerated Gradient Descent (AGD) Algorithm

---

1: **Input:** training data $\mathcal{D} = \{(\boldsymbol{x}_n, y_n)\}_{n=1}^N$, gradient step size $\eta$, momentum parameter $\beta$, and an initialization $\boldsymbol{W}^{(0)}$ by the tensor initialization method;
2: Partition $\mathcal{D}$ into $T = \log(1/\varepsilon)$ disjoint subsets, denoted as $\{\mathcal{D}_t\}_{t=1}^T$;
3: **for** $t = 1, 2, \cdots, T$ **do**
4:     $\boldsymbol{W}^{(t+1)} = \boldsymbol{W}^{(t)} - \eta \nabla_{\boldsymbol{W}} \hat{f}_{\mathcal{D}_t}(\boldsymbol{W}^{(t)}) + \beta(\boldsymbol{W}^{(t)} - \boldsymbol{W}^{(t-1)})$
5: **end for**
6: **Return:** $\boldsymbol{W}^{(T)}$

---

1. **Linear convergence to the ground-truth model**: Theorem 2 implies that if initialized in the local convex region, the iterates generated by AGD converge linearly to the ground truth $\boldsymbol{W}^*$ when noiseless. When there is noise, they converge to a point $\boldsymbol{W}^{(T)}$. The distance between $\boldsymbol{W}^{(T)}$ and $\boldsymbol{W}^*$ is proportional to the noise level and scales in terms of $\mathcal{O}(\sqrt{r/N})$. Moreover, when $N$ is fixed, the convergence rate of AGD is $\Theta(\sqrt{r/K})$. Recall that Algorithm 1 reduces to the vanilla GD by setting $\beta = 0$. The rate for the vanilla GD algorithm here is $\Theta(\sqrt{r/K})$ by setting $\beta = 0$ by Theorem 2, indicating a slower convergence than AGD. Lemma 1 shows the tensor initialization method indeed returns an initial point in the convex region.

2. **Sample complexity for accurate estimation**: We show that the required number of samples for successful estimation is $\Theta(r \log d)$, which is order-wise optimal with respect to the number of non-pruned neuron weights $r$, and only logarithmic with respect to the input feature dimension $d$. Our sample complexity is much less than the conventional bound of $\Theta(d \log d)$ for one-hidden-layer networks (Zhong et al., 2017; Zhang et al., 2020a;c). This is the first theoretical characterization of learning a pruned network from the perspective of sample complexity.

3. **Improved generalization of winning tickets**: We prove that with a fixed number of training samples, training on a properly pruned sub-network converges faster to $\boldsymbol{W}^*$ than training on the original dense network. Our theoretical analysis justifies that training on the winning ticket can meet or exceed the same test accuracy within the same number of iterations. To the best of our knowledge, our result here provides the first theoretical justification for this intriguing empirical finding of "improved generalization of winning tickets" by (Frankle & Carbin, 2019).

**Theorem 2** (Convergence). *Suppose $\boldsymbol{W}^{(0)}$ satisfies (6) and the number of samples satisfies*

$$N = \Omega\big(\varepsilon_0^{-2} K^8 r \log d \log(1/\varepsilon)\big) \tag{8}$$

*for some $\varepsilon_0 \in (0, 1/2)$. Let $\eta = K/14$ in Algorithm 1. Then the iterates $\{\boldsymbol{W}^{(t)}\}_{t=1}^T$ returned by Algorithm 1 converges linearly to $\boldsymbol{W}^*$ up to the noise level with probability at least $1 - K^2 T \cdot d^{-10}$ as*

$$\|\boldsymbol{W}^{(t)} - \boldsymbol{W}^*\|_F \le \nu(\beta)^t \|\boldsymbol{W}^{(0)} - \boldsymbol{W}^*\|_F + \mathcal{O}\big(\sqrt{K^2 r \log d/N}\big) \cdot |\xi|, \tag{9}$$

$$\|\boldsymbol{W}^{(t)} - \boldsymbol{W}^*\|_\infty \le \nu(\beta)^t \|\boldsymbol{W}^{(0)} - \boldsymbol{W}^*\|_\infty + \mathcal{O}\big(\sqrt{K^2 r \log d/N}\big) \cdot |\xi|, \tag{10}$$

$$\|\boldsymbol{W}^{(T)} - \boldsymbol{W}^*\|_F \le \varepsilon \|\boldsymbol{W}^*\|_F + \mathcal{O}\big(\sqrt{K^2 r \log d/N}\big) \cdot |\xi|, \tag{11}$$

*where $\nu(\beta)$ is the rate of convergence that depends on $\beta$ with $\nu(\beta^*) = 1 - \Theta\big(\frac{1-\varepsilon_0}{\sqrt{K}}\big)$ for some non-zero $\beta^*$ and $\nu(0) = 1 - \Theta\big(\frac{1-\varepsilon_0}{K}\big)$.*

**Lemma 1** (Initialization). *Assume the noise $|\xi| \le \|\boldsymbol{W}^*\|_2$ and the number of samples $N = \Omega\big(\varepsilon_0^{-2} K^8 r \log^4 d\big)$ for $\varepsilon_0 > 0$, the tensor initialization method outputs $\boldsymbol{W}^{(0)}$ such that (6) holds, i.e., $\|\boldsymbol{W}^{(0)} - \boldsymbol{W}^*\|_F = \mathcal{O}\big(\frac{\varepsilon_0 \sigma_K}{K^2}\big)$.*

With a fixed number of samples, when $r$ decreases, $\varepsilon_0$ can be $\Theta(\sqrt{r})$ while the condition in (8) is still met. Then $\nu(0) = \Theta(\sqrt{r}/K)$ and $\nu(\beta^*) = \Theta(\sqrt{r/K})$. Therefore, when $r$ decreases, both the stochastic and accelerated gradient descent converge faster. The theoretical bound of the improvement of the convergence rate by accelerated gradient descent is the same as that in (Zhang et al., 2020a;c). However, (Zhang et al., 2020a;c) focus on convolutional neural networks without any network pruning, while our results consider pruned networks. Note that as long as $\boldsymbol{W}^{(0)}$ is initialized in the local convex region, not necessarily by the tensor method, Theorem 2 guarantees the accurate recovery.

In our proof of Theorem 2, we need to address the technical challenge that does not appear in (Zhong et al., 2017) such that each $\boldsymbol{w}_j$ connects to a different subset of $\boldsymbol{x}$ here instead of the same $\boldsymbol{x}$ in (Zhong et al., 2017). Hence, the concentration theorem cannot be directly applied here to bound the distance between population and empirical risk function as used in (Zhong et al., 2017). Moreover, we need to revise the tensor initialization method and the corresponding proof due to the pruned network architecture. If we choose $r = d$, our analysis reduces to the case in (Zhong et al., 2017). Moreover, our algorithm enjoys a faster convergence rate since we consider AGD method rather than GD as in (Zhong et al., 2017).

## 4.1 Obtaining a proper student network via magnitude pruning

We next show that one can combine Algorithm 1 and magnitude pruning to find a proper student network such that $r \geq r^*$ and $\Omega_j \supseteq \Omega_j^*$ from a fully-connected network under some assumptions. Suppose the number of samples is at least $\Omega\big(K^8 d \log d \log(1/\varepsilon)\big)$, we train directly on the fully-connected dense network using Algorithm 1. The number of iteration in line 2 of Algorithm 1 is set as $T_1 = \Theta\big(\log(2\widehat{W}_{\max}/\widehat{W}_{\min})\big)$, where $\widehat{W}_{\min}$ and $\widehat{W}_{\max}$ denote the smallest and largest value of $\boldsymbol{W}^*$, respectively. From (10), after $T_1$ iterations, the returned model, denote by $\boldsymbol{W}^{(T_1)}$, is close to the augmented ground-truth $\widetilde{\boldsymbol{W}}^*$. Specifically, if $\widetilde{W}_{ij}^* \neq 0$ and $\widetilde{W}_{i'j'}^* = 0$, then $\boldsymbol{W}_{ij}^{(T_1)} > \boldsymbol{W}_{i'j'}^{(T_1)}$ for any $i, j, i', j'$ (See Appendix-E for details). Then we sort the weights based on their absolute values and prune them sequentially starting from the least absolute value. As long as the ratio of pruned weights is at most $1 - r^*/d$, all the weights are removed correctly, leading to a proper student network. In fact, if we remove exactly $1 - r^*/d$ fraction of weights, the pruned network has the same architecture as the teacher network.

## 5 Numerical Experiments

We first verify our theoretical results on synthetic data and then analyze the pruning performance of the IMP algorithm (Frankle & Carbin, 2019) on both synthetic data and real data. The synthetic data are generated using a ground-truth sparse neural network in Fig. 1. The input $\{\boldsymbol{x}_n\}_{n=1}^N$ are randomly generated from Gaussian distribution $\mathcal{N}(0, \boldsymbol{I}_{d \times d})$ independently, and the index set $\Omega_j(1 \leq j \leq K)$ is generated by selecting $r$ numbers randomly from $[d]$ without replacement. Each entry of the weight matrix $\boldsymbol{W}^*$ is randomly selected from $[-0.5, 0.5]$ independently. The noise $\{\xi_n\}_{n=1}^N$ are i.i.d. from $\mathcal{N}(0, \sigma^2)$, and the noise level is measured by $\sigma/E_y$, where $E_y$ is the average energy of the noiseless outputs $\{g(\boldsymbol{x}_n; \boldsymbol{W}^*)\}_{n=1}^N$ calculated as $E_y = \sqrt{\frac{1}{N}\sum_{n=1}^N |g(\boldsymbol{x}_n; \boldsymbol{W}^*)|^2}$.

Algorithm 1 is implemented with two minor modifications. First, the initial point is randomly selected from $\big\{\boldsymbol{W}^{(0)}\big| \|\boldsymbol{W}^{(0)} - \boldsymbol{W}^*\|_F / \|\boldsymbol{W}^*\|_F < \lambda\big\}$ for some constant $\lambda > 0$ to reduce the computation. Second, we use the whole training data instead of a fresh subset in each iteration. Algorithm 1 terminates when $\|\boldsymbol{W}^{(t+1)} - \boldsymbol{W}^{(t)}\|_F / \|\boldsymbol{W}^{(t)}\|_F$ is smaller than $10^{-8}$ or reaching 10000 number of iterations. In Section 5.2,

we implement the IMP algorithm (Frankle & Carbin, 2019) [4] to prune the neural network. The real dataset we use is the MNIST dataset, and the network architecture is Lenet-5 structure (Lecun et al., 1998).

## 5.1 EVALUATION OF THEORETICAL FINDINGS ON SYNTHETIC DATA

**Local convexity near the ground-truth.** We set the number of neurons $K = 5$, the dimension of the data $d = 500$ and the sample size $N = 3000$. Figure 2 illustrates the success rate of Algorithm 1 when the network sparsity changes. The $y$-axis is the relative distance of the initialization $\boldsymbol{W}^{(0)}$ to the ground-truth. For each pair of $r$ and the initial distance, we run 100 independent tests, and the network weights, training data and the initialization $\boldsymbol{W}^{(0)}$ are all generated independently in each test. Each test is called successful if the relative error of the solution $\boldsymbol{W}$ returned by Algorithm 1, measured by $\|\boldsymbol{W} - \boldsymbol{W}^*\|_2/\|\boldsymbol{W}^*\|_2$, is less than $10^{-4}$. A black block means Algorithm 1 fails in estimating $\boldsymbol{W}^*$ in all runs while a white block indicates all successes. Algorithm 1 succeeds if $\boldsymbol{W}^{(0)}$ is in the local convex region near $\boldsymbol{W}^*$. From Figure 2, we can see that the radius of convex region is indeed linear in $-\sqrt{r}$, as predicted by Theorem 1.

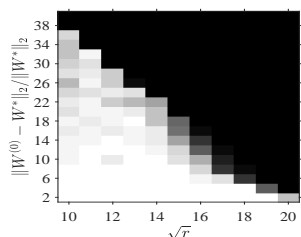

Figure 2: The radius of the local convex region against $\sqrt{r}$

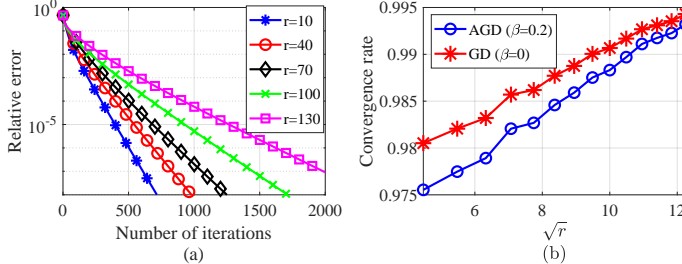

Figure 3: Convergence rate when $r$ changes

**Convergence rate.** Figure 3 shows the convergence rate of Algorithm 1 when $r$ changes. $N = 3000$, $d = 300$, $K = 5$, $\eta = 0.5$, and $\beta = 0.2$. Figure 3(a) shows that the relative error decreases exponentially as the number of iterations increases, indicating the linear convergence of Algorithm 1. As shown in Figure 3(b), the convergence rate is almost linear in $\sqrt{r}$, as predicted by Theorem 2. We also compare with GD by setting $\beta$ as 0. One can see that AGD has a smaller convergence rate than GD, indicating faster convergence.

**Sample complexity.** Figure 4 shows the phrase transition of Algorithm 1 when varying $N$ and $r$. $d$ is fixed as 100. For each value of $N$, we construct 100 independent runs where the ground-truth model and training data are generated independently in each run. We can see that the required number of samples for successful estimation is linear in $r$, as predicted by (8).

**Performance in noisy case.** Figure 5 shows the relative error of the learned model by Algorithm 1 from noisy measurements when $r$ changes. $N = 1000$, $K = 10$, and $d = 300$. The relative error is linear in $\sqrt{r}$, as predicted by (9). Moreover, the relative error is proportional to the noise level $|\xi|$.

## 5.2 IMP FOR FINDING WINNING TICKETS

Here we implement the IMP algorithm to obtain pruned networks on both synthetic data and real data. Figure 6 shows the test performance of a pruned network on synthetic data with different sample sizes. Here in the ground-truth network model, $K = 5, d = 100$, and $r/d = 20\%$. The noise level $\sigma/E_y = 10^{-3}$. One observation is that for a fixed sample size $N$ greater than 100, the test error decreases as the number of remaining parameters decreases. This verifies that the IMP algorithm indeed prunes the network properly. It also shows that the learned model improves as the pruning progresses, verifying our theoretical result in

---

[4]The code is downloaded from https://github.com/rahulvigneswaran/Lottery-Ticket-Hypothesis-in-Pytorch.

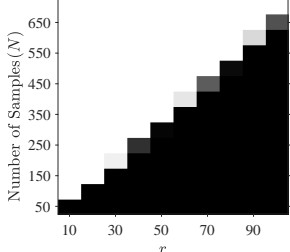 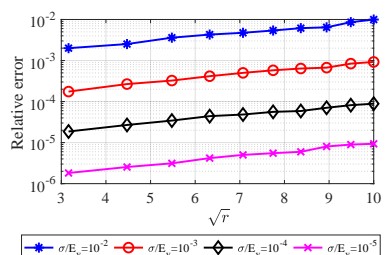

Figure 4: Sample complexity when $r$ changes    Figure 5: Relative error against $\sqrt{r}$ at different noise level

Theorem 2 that the difference of the learned model from the ground-truth model decreases as the number of remaining weights decreases. The second observation is that the test error decreases as $N$ increases for any fixed number of remaining parameters. This verifies our result in Theorem 2 that the difference of the learned model from the ground-truth model decreases as the number of training samples increases. When the network is pruned significantly such that the percentage of reaming parameters is less than the ground-truth 20%, the pruned network cannot explain the data properly, and thus the test error is large for all $N$. When the number of samples is too small, $N = 100$, the test error is always large, because it does not meet the sample complexity requirement for estimating the sparse model even though the network is properly pruned.

Figure 7 shows the performance of the IMP algorithm on MNIST dataset using Lenet5 architecture. The percentage of weights to be pruned after each cycle is 20%, and other parameters are set as default values. Although not as obvious as the results on synthetic data, we still observe the same phenomenon of the performance when pruning progresses. That is, the test accuracy first increases when we properly prune the network (e.g., the case of $N = 20000$), indicating the effectiveness of proper pruning. By contrast, the test accuracy drops as the network is overly pruned.

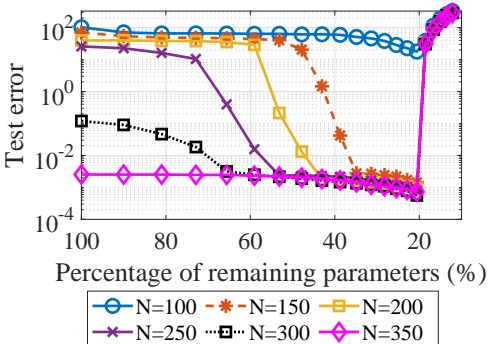 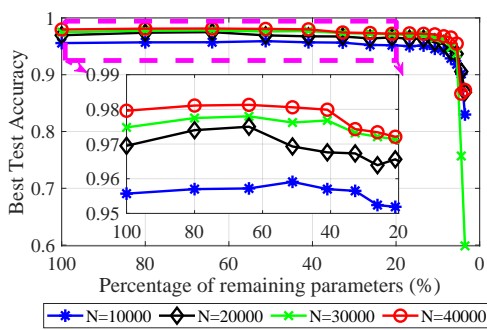

Figure 6: Test error of pruned models on the synthetic dataset    Figure 7: Test accuracy of pruned models on MNIST dataset

## 6   CONCLUSIONS

This paper provides the first theoretical analysis of learning one-hidden-layer sparse neural networks, which offers formal justification of the improved generalization of winning ticket observed from empirical findings in LTH. We characterize analytically the impact of the number of remaining weights in a pruned network on the required number of samples for training, the convergence rate of learning algorithm, and the accuracy of the learned model. We also provide extensive numerical validations of our theoretical findings. One desired future work will be generalizing our theoretical analysis to the scenario of network pruning on multi-layer neural networks.

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
