# OpenReview forum: "Why Lottery Ticket Wins? A Theoretical Perspective of Sample Complexity on Sparse Neural Networks"
_ICLR.cc/2021/Conference — Reject_

### Official Review · AnonReviewer1 · 2020-10-28
**Review of Why Lottery Ticket Wins? A Theoretical Perspective of Sample Complexity on Sparse Neural Networks**

**Rating:** 7
**Confidence:** 4

**Review:**

Summary:

This paper investigates the theoretical evidence behind improved generalization of the winning lottery tickets. More precisely authors characterize the testing error of a pruned network that is then trained using AGD. Under relatively reasonable assumptions, they manage to show an improved generalization bound for a properly pruned network over the full network.

#######################################################################
pros:

This is one of the first works to provide theoretical guarantees for winning lottery tickets under the one-hidden-layer sparse neural network model. The objective function being highly non-convex in general, authors rely on local convexity of the latter objective function around the ground truth. Hence Given a good initialization, they managed to show that AGD achieves a good performance after a certain number of steps. Numerical experiments are interesting and complement pretty well the theory of the paper.

#######################################################################
cons/questions:

* Line 6 page 5, you say that the radius of the convex ball is $\Theta(1) - \Theta(\sqrt{r})$. How is that even positive as $r$ grows? The same comment also holds for your comment after Lemma 1.

*  Using local convexity, which is a uniform property, you should be able to prove your main result based on the whole training data instead of sample splitting. I don't see why you have only managed to show your result under sample splitting which is less interesting in my opinion.

* The hypothesis of a sparse ground model where you are given the true corresponding support is too restrictive. It would have been very interesting to apply a variant of IHT to your proposed algorithm. I believe your results should hold under relatively similar conditions. I strongly think proving your result without prior knowledge of the true support would make the present paper stronger and would avoid having trivial follow-up works that may take more credit than yours.

#######################################################################
 Score:

This paper is well written and the proofs seem sound to me. Overall, I think the present paper is marginally above the acceptance threshold because of the reasons I explain above. I am willing to revise my score if the authors give constructive feedback to my concerns.

Comments:

* In Theorem 7, it would be good to precise that your initialization is independent from the data you use for your AGD.

---

> ### Author Response · Authors · 2020-11-21
> **Summary of the changes for the paper**
>
> Thank you for the comments and suggestions. Here are some major revisions in the papers:
>
> $\bullet$ We have revised the problem formulation to clarify our setup.
> Our setup can be interpreted by a "teacher-student" mode,
> where the training data are generated from a teacher network, and we learn on a student network with the same number of neurons as the teacher network. We do  not require the student network to have the same architecture as the teacher network. As long as the teacher network can be viewed as a sub-network of the student network by pruning weights, our method finds the ground-truth model with a sufficient number of samples.  We study how the architecture of the student network affects the learning rate and sample complexity in this paper.
>
> $\bullet$ Following the reviewers comments, we added one section (Section 4.1) to discuss the guarantee of obtaining a proper student network architecture from a fully connected dense network by magnitude pruning. The main idea is that after some iterations of learning on the dense network using Algorithm 1, the weights that correspond to a non-existing edge in the teacher network becomes sufficient small (in magnitude), then magnitude pruning removes edges correctly.

---

> > ### Author Response · Authors · 2020-11-21
> > **Replies to the comments**
> >
> > The point to point responses to the comments are summarized in the following:
> >
> > $\bullet$ Q1: Line 6 page 5, you say that the radius of the convex ball is $\Theta(1) - \Theta(\sqrt{r})$. How is that even positive as  grows? The same comment also holds for your comment after Lemma 1.
> >
> > $\bullet$ A1: Sorry for the confusion. To present it more accurately, we have revised corresponding notation to $\Theta(\sqrt{r/N})$.
> > 	With $\Theta(1)-\Theta(\sqrt{{r/N}})$, we want to emphasize that there exists some contents $c_1$ and $c_2$ such that the radius is positive and given by $c_1 - c_2\cdot \sqrt{r/N}$.
> >
> > $\bullet$ Q2: Using local convexity, which is a uniform property, you should be able to prove your main result based on the whole training data instead of sample splitting. I don't see why you have only managed to show your result under sample splitting which is less interesting in my opinion.
> >
> > $\bullet$ A2: Thanks for the comments. The reason we still need to split the samples is because the empirical risk function is non-smooth since we consider ReLU function here. To the best of our knowledge, only (Fu et al, ``Guaranteed recovery of one-hidden-layer neural networks via cross entropy'', 2018) used the whole training data instead of sample splitting, however, the activation function is sigmoid in (Fu et al, 2018) and the mean-vaue theorem can be applied.  As the empirical risk function is no longer smooth in our setup, we cannot directly apply the mean-value theorem used in (Fu et al, 2018). Hence, even if we can prove the uniformly local convexity  property,
> > we still need the sample splitting for the convergence analysis. Meanwhile, the sample splitting technique does not degenerate the theoretical results too much as it only adds a factor of $\log(1/\varepsilon)$, which can be viewed as a constant in most cases.
> > here may be other ways to avoid sample splitting for non-smooth activation function. We are still currently working on it and will update if we manage to solve it.
> > $\bullet$ Q3: The hypothesis of a sparse ground model is too restrictive. It would have been very interesting to apply a variant of IMP to your proposed algorithm. I believe your results should hold under relatively similar conditions. I strongly think proving your result without prior knowledge of the true support would make the present paper stronger and would avoid having trivial follow-up works that may take more credit than yours.
> >
> > $\bullet$  A3: Thank you for the   enlightening suggestion. We revised our description of the setup to clarity that we do not need the ground-truth model to be sparse. We also do not need to know the ground-truth architecture. Moreover, we added a subsection 4.1 to discuss the guarantee to find a proper architecture using magnitude pruning from a fully connected network. Details as as follows.
> >
> > First, we need to clarify that we consider a teacher-student setup where the data are generated from a teacher network with $r^*$ weights per neuron. We learn on a student network with $r$ weights per neuron. We only assume $r^* \leq r \leq d$ throughout the paper, and $r^*$ and $r$ can take the entire spectrum of weight density, from sparse to dense network. Thus, our formulation is associated with a quite general case.
> > We removed the statement about sparsity that could lead to confusion.
> >
> > Second, the student network does not need to be the same as the teacher network. As long as the teacher network is a subnetwork of the student network, our results in Theorems 1 and 2  applies. In other words, we do not assume knowing the mask of the teacher network exactly.
> >
> > Third, we added a subsection 4.1 to discuss how to obtain a proper student network from a fully connected neural network using our Algorithm 1 and magnitude pruning. We can train on a fully connected neural network using tensor initialization and iterate with $T_1$ iterations, then apply magnitude pruning to prune the network. We show that the teacher network is indeed a subnetwork of the resulting pruned network. Then this pruned network can be used as the student network discussed in this paper. Please see section 4.1 in the paper for details.

---

> > > ### Comment · AnonReviewer1 · 2020-11-23
> > > **Replies to the revision**
> > >
> > > I would like the authors for taking the time to address my concerns.
> > >
> > > For me, the main problem in this paper is in the hypothesis of a sparse ground model where the support is known. I can see that the authors tried to deal with this problem. In their solution, authors suggest to first use IMP (section 4.1) to recover a solution that contains the ground truth weights and then apply the original AGD. Overall this two steps approach requires a sample size of order $d\log(1/\epsilon)$ unlike what we would expect for sparse sub-network estimation that is given by $r\log(1/\epsilon)$.
> > >
> > > While I can see why this new condition on the sample size is required for IMP to work, I am not satisfied with. I still think that a good initialization followed by an iterative hard thresholding procedure should be enough to get the final result under the requirement $N$ of order $r\log(1/\epsilon)$.
> > >
> > > For these reasons,  I will maintain my score unchanged for now and encourage the authors to think a bit more about this issue.

---

> > > > ### Author Response · Authors · 2020-11-24
> > > > **Replies to the comment**
> > > >
> > > > Thank you very much for your response to our revision. We now understand your point better. Although we agree with the reviewer that, intuitively, a good initialization followed by an iterative hard thresholding (IHT) procedure may provide a better model, but such this intuition is not easy to formalize theoretically due to the non-convexity of the learning problem here.
> > > >
> > > > Since the reviewer mentioned IHT, a standard terminology in compressed sensing (CS), we would like to highlight the difference in analyzing model compression compared to CS. Yes, one can estimate a $k$-sparse signal from $O(k\log n)$ measurements without knowing the support of the signal in CS. However, compared to model compression, the method in CS, e.g., the well-known $\ell_1$-minimization, is a convex problem, which enables the global analysis.
> > > >
> > > > In our case, to show that the model improves in each iteration in the iterative approach, we will need to show the gradient descent steps in each iteration indeed moves towards the ground-truth model. That in turns requires local convexity. Without the local convexity, one cannot show that using gradient descent steps on the current architecture can provide a better model. However, as indicated by Theorem 1 and the discussion after it, when the number of samples is fixed, the radius of the local convex region reduces as $r$ increases. Thus, if $N=O(r\log 1/\epsilon)$ and we train on a network starting with $r=d$, the local convex region is very small. Then if initialized outside this region (most likely the case), there is no guarantee that gradient descent would improve the model, although intuitively we think it would. Therefore, the bound $O(d\log (1/\epsilon)$ in Section 4.1 leads to a larger convex region so that we could analyze, although we agree it might not be tight.
> > > >
> > > > We agree that in practice, a good initialization followed by the IHT procedure may work well, but the limited tools in nonconvex optimization prevent us from quantifying that. We believe it would require the development of new tools to enable such analysis. This is definitely an interesting future direction that we would like to work on. Thanks!

---

### Official Review · AnonReviewer2 · 2020-10-29

**Rating:** 5
**Confidence:** 2

**Review:**

The paper analyzes the geometric structure of the objective function for a sparse one hidden layer neural net, and has a novel theorem on the convergence rate of the algorithm that recovers the weights in the one hidden layer neural net. From the perspective of sample complexity, namely how many samples are needed to have the whole recovery of the weights in the neural net, it is established that a sparse one hidden layer neural net usually requires fewer samples than a fully connected counterpart. The paper uses many simulations to support the theoretical results. The sparse neural net can represent the winning ticket for the lottery ticket hypothesis, and the sample complexity explains why the winning ticket has better performance. The paper seems the first work to focus on the weights recovery of sparse neural networks and provides useful guarantees with important insights, and the paper should be distributed to other researchers.

Although the paper has many contributions and should be worth distributed to other researchers, the paper appears to be just below the threshold for acceptance to the ICLR conference. Therefore, I am afraid I might not recommend to accept the paper in this current form.

1. Frankly, the technical novelty over the reference Zhong 2017 (Recovery guarantees for one-hidden-layer neural networks) seems not fully clarified. While it is accepted that this work has the sparsity $r$ and proves a tighter bound than Zhong 2017, it seems not fully clarified what technical novelty or key breakthrough are directly due to the sparsity setting in this paper, compared with the approach that Zhong 2017 took. The authors should highlight key technical differences in the proof due to the sparsity setting, to make it more convincing about the novelty of this paper.
2. The paper only explains why a sparse neural net should train well with fewer examples than fully connected counterpart. The observation has its own merit. To my understanding, the paper starts from the assumption that the neural net to be learned is sparse – however, this could not be always correct for the lottery ticket problem. Thus, the assumption seems not always plausible for real applications of the lottery ticket hypothesis, and it would be useful if there can be examples of sparse neural networks that come naturally for some problem.
3. Here is a question for the authors to confirm. Although the work is on sparse neural networks, the sparsity $r<<d$ is not assumed in Theorem 1 and 2, correct? Even if $r=d$, we still have the theorems valid? Please feel free to let me know if I do not understand well.

---

> ### Author Response · Authors · 2020-11-21
> **Summary of the changes for the paper, and the replies to the comments.**
>
> Thank you for the comments and suggestions. Here are some major revisions in the papers:
>
> $\bullet$ We have revised the problem formulation to clarify our setup.
> Our setup can be interpreted by a "teacher-student" mode,
> where the training data are generated from a teacher network, and we learn on a student network with the same number of neurons as the teacher network. We do  not require the student network to have the same architecture as the teacher network. As long as the teacher network can be viewed as a sub-network of the student network by pruning weights, our method finds the ground-truth model with a sufficient number of samples.  We study how the architecture of the student network affects the learning rate and sample complexity in this paper.
>
> $\bullet$ Following the reviewers comments, we added one section (Section 4.1) to discuss the guarantee of obtaining a proper student network architecture from a fully connected dense network by magnitude pruning. The main idea is that after some iterations of learning on the dense network using Algorithm 1, the weights that correspond to a non-existing edge in the teacher network becomes sufficient small (in magnitude), then magnitude pruning removes edges correctly.
>
> The point to point responses to the comments are summarized in the following:
>
> $\bullet$ Q1: The technical novelty over the reference Zhong 2017 (Recovery guarantees for one-hidden-layer neural networks) seems not fully clarified.
>
> $\bullet$ A1: Thanks for the comments, and we are sorry that the differences with Zhong etal'17  were not well presented.
> The main differences of our proof from Zhong et al'17 lie in two aspects:
>
> First, each neuron weights $w_j$ connects to different subset of $x$ instead of fixed $x$. Hence, the concentration theorem cannot be directly applied in this paper to bound the distance between population and empirical risk function.
>
> Second, we need to replace $x$ with $\frac{1}{\sqrt{K}}\sum_{j=1}^K x_j$ for the tensor initialization since $w_j$ is no longer respect with $x$.
> Although the follow the road map of the proof  in Zhong et al'17, the proof techniques to guarantee the lemmas still hold are totally different because of the difference we described above.
>
> We have added a separate paragraph after Theorem 2 to clarify the differences with Zhong et al'17.
>
> $\bullet$ Q2: To my understanding, the paper starts from the assumption that the neural net to be learned is sparse - however, this could not be always correct for the lottery ticket problem. Thus, the assumption seems not always plausible for real applications of the lottery ticket hypothesis, and it would be useful if there can be examples of sparse neural networks that come naturally for some problem.
>
> $\bullet$ A2: Thanks for the comments.
> We do not require the learned neural net to be sparse, and we are sorry our original statement was confusing.
>
> First, we need to clarify that we consider a teacher-student setup where the data are generated from a teacher network with $r^*$ weights per neuron. We learn on a student network with $r$ weights per neuron. We only assume $r^* \leq r \leq d$ throughout the paper, and $r^*$ and $r$ can take the entire spectrum of weight density, from sparse to dense network. Thus, our formulation is associated with a quite general case. We removed the statement about sparsity that could lead to confusion.
>
> Second, the student network does not need to be the same as the teacher network. As long as the teacher network is a subnetwork of the student network, our results in Theorems 1 and 2  applies. In other words, we do not assume knowing the mask of the teacher network exactly.
>
> Third, our work provides an explicit characterization of the impact of pruning on the sample complexity and the learning rate. The sample complexity decreases, and the learning becomes faster when $r$ in the student network decreases. If the teacher network has a small $r^*$, one can find a proper student network (using our approach in the new added Section 4.1) with a small $r$ such that the learning performance on the student network is much improved over training on a fully connected network.
>
> $\bullet$ Q3: Although the work is on sparse neural networks, the sparsity  is not assumed in Theorem 1 and 2, correct?
>
> $\bullet$ A3: Thanks for bringing up the confusion. As our response in Q2, the sparsity is not required throughout the paper, and the theorems   hold  for any $r\le d$. We apologize for the misunderstanding sentence in the original paper.

---

### Official Review · AnonReviewer3 · 2020-11-02
**Theoretical results for learning a one-hidden-layer neural network with sparse ground truth weights given a Gaussian input distribution**

**Rating:** 5
**Confidence:** 4

**Review:**

Summary of review:

This paper provides recovery guarantees for learning one-hidden-layer neural networks with sparse ground truth weights, given an isotropic Gaussian input distribution. The main result shows local convexity guarantees near the ground truth. Provided that a mask of the sparsity pattern is already known, this paper extends the tensor initialization approach of Zhong et al'17 to show a convergence guarantee for learning the sparse neural network. Simulations validate the local

Setting:

This paper focuses on learning a one-hidden-layer neural network, where the weight matrix of the hidden layer is sparse, given input samples from an isotropic Gaussian distribution.

Results:

(i) The first result is that within a small vicinity of the ground truth weight matrix, the standard mean squared loss for learning the neural network is convex.

(ii) The second result shows how to learn the ground truth weight matrix, by extending the tensor initialization approach in Zhong et al'17.

(iii) Numerical results are provided to validate the above two theoretical results.

Pros:

- The sample size requirement of both result (i) and (ii) growly proportionally to the sparsity of the ground truth matrix, as opposed to the size of the matrix. This result is particularly interesting in light of recent empirical results about network pruning and learning sparse ConvNets (Neyshabur'20).

Cons:

- The authors prove the above results by adapting the proof of Zhong et al'17. In fact, since the input distribution is isotropic, standard concentration results apply whether or not the ground truth matrix is sparse. Therefore, it is unclear to the reviewer whether this result is as novel as the authors claim in the introduction.

- The learning algorithm assumes knowledge of the sparsity mask. This seems like a strong assumption. Isn't the point of IMP to find this sparsity mask? Understanding how to find this sparsity mask seems like a more important question, but this is not discussed at all in this paper.

Writing:

Overall, this paper is easy to follow. The quality of writing is marginally acceptable. Please find several detailed comments below.

- P1, "the theoretical justification of winning tickets are remains elusive expect for a few recent works" -> remove "are", replace "expect" with except

- P4: "an one-hidden-layer neural network" -> a one-hidden-layer neural network

- P5: here you say that $\varepsilon_1 = \Theta(\sqrt r)$, but $r > 1$ and $\varepsilon_1 < 1$. Please clarify.

- P5: regarding the convergence for the vanilla GD algorithm. Please add a reference to this claim.

- P5: "accurate estimate" -> replace "estimate" with estimation

---

> ### Author Response · Authors · 2020-11-21
> **Summary of the changes for the paper, and the replies to the comments in "Cons".**
>
> Thank you for the comments and suggestions. Here are some major revisions in the papers:
>
> $\bullet$ We have revised the problem formulation to clarify our setup.
> Our setup can be interpreted by a "teacher-student" mode, where the training data are generated from a teacher network, and we learn on a student network with the same number of neurons as the teacher network. We do  not require the student network to have the same architecture as the teacher network. As long as the teacher network can be viewed as a sub-network of the student network by pruning weights, our method finds the ground-truth model with a sufficient number of samples.  We study how the architecture of the student network affects the learning rate and sample complexity in this paper.
>
> $\bullet$ Following the reviewers comments, we added one section (Section 4.1) to discuss the guarantee of obtaining a proper student network architecture from a fully connected dense network by magnitude pruning. The main idea is that after some iterations of learning on the dense network using Algorithm 1, the weights that correspond to a non-existing edge in the teacher network becomes sufficient small (in magnitude), then magnitude pruning removes edges correctly.
>
> The point to point responses to the comments are summarized in the following:
>
> $\bullet$	Q1: The authors prove the above results by adapting the proof of Zhong et al'17. In fact, since the input distribution is isotropic, standard concentration results apply whether or not the ground truth matrix is sparse. Therefore, it is unclear to the reviewer whether this result is as novel as the authors claim in the introduction.
>
>
> $\bullet$ A1: Thanks for the comments, and we are sorry that the differences with Zhong etal'17  were not well presented. However, we do not think the statement "since the input distribution is isotropic, standard concentration results apply whether or not the ground truth matrix is sparse'' is correct.
> The main differences of our proof from Zhong et al'17 lie in two aspects:
>
> First, each neuron weights $w_j$ connects to different subset of $x$ instead of fixed $x$. Hence, the concentration theorem cannot be directly applied in this paper to bound the distance between population and empirical risk function.
>
> Second, we need to replace $x$ with $\frac{1}{\sqrt{K}}\sum_{j=1}^K x_j$ for the tensor initialization since $w_j$ is no longer respect with $x$. % We do not deny that the proof shares similarity with Zhong et al's in presenting theorems and lemmas because the roadmap of this paper follows exactly the same as that in Zhong et al's. However,
> 		Although the follow the road map of the proof  in Zhong et al'17, the proof techniques to guarantee the lemmas still hold are totally different because of the difference we described above.
>
> We have added a separate paragraph after Theorem 2 to clarify the differences with Zhong et al'17.
>
> $\bullet$	Q2: The learning algorithm assumes knowledge of the sparsity mask. This seems like a strong assumption. Isn't the point of IMP to find this sparsity mask? Understanding how to find this sparsity mask seems like a more important question, but this is not discussed at all in this paper.
>
> $\bullet$	A2: Thanks for bringing the question. We realize that our setup was not discussed accurately and thus may lead  to confusions.  We need to clarify that our algorithm does not need to know the mask accurately. Moreover, we added a subsection 4.1 to discuss the guarantee to find a mask that works for our algorithm using magnitude pruning from a dense networks. Details as as follows.
>
>
> First, we need to clarify that we consider a teacher-student setup where the data are generated from a teacher network with $r^*$ weights per neuron. We learn on a student network with $r$ weights per neuron. We only assume $r^* \leq r \leq d$ throughout the paper, and $r^*$ and $r$ can take the entire spectrum of weight density, from sparse to dense network. Thus, our formulation is associated with a quite general case.
> We removed the statement about sparsity that could lead to confusion.
>
> Second, the student network does not need to be the same as the teacher network. As long as the teacher network is a subnetwork of the student network, our results in Theorems 1 and 2  applies. In other words, we do not assume knowing the mask of the teacher network exactly.
>
> Third, we added a subsection 4.1 to discuss how to obtain a proper student network from a fully connected neural network using our Algorithm 1 and magnitude pruning. We can train on a fully connected neural network using tensor initialization and iterate with $T_1$ iterations, then apply magnitude pruning to prune the network. We show that the teacher network is indeed a subnetwork of the resulting pruned network. Then this pruned network can be used as the student network discussed in this paper. Please see section 4.1 in the paper for details.

---

> > ### Author Response · Authors · 2020-11-21
> > **Replies to the comments in "Writing"**
> >
> > $\bullet$ Q3: writing
> >
> > $\bullet$ A3: For the writing part, thank you for pointing out the some grammar issues.
> >
> > For question ``here you say that $\varepsilon_1 = \Theta(\sqrt r)$, but $r>1$ and $\varepsilon_1<1$.  Please clarify.", sorry for the confusion. To present it more accurately, we have revised corresponding notation to $\Theta(\sqrt{r/N})$. $\varepsilon_1 = \Theta(\sqrt {r/N})$ stands for there exists a constant $c_1$ and $c_2$ such that $c_1 \sqrt{{r/N}} \le\varepsilon_1 \le   c_2 \sqrt{{r/N}} $, and the constant $c_1$ and $c_2$ can be small enough to guarantee $\varepsilon_1<1$. The notation means that $\varepsilon_1$ increases as a linear function of $\sqrt{{r/N}}$.
> >
> > For question ``P5: regarding the convergence for the vanilla GD algorithm. Please add a reference to this claim."
> > We apologize for the confusion. The rate for GD is derived from Theorem 2 by setting $\beta = 0$.  If $\beta = 0$, AGD is reduced to GD, and we show that the convergence rate is $\Theta(\sqrt{r/K})$. Although GD has been studied in fully connected neural networks, for instance,  (Zhong et al., 2017).   shows that its convergence rate is $\Theta(\sqrt{d/K})$. No existing works have discussed the rate of GD in pruned networks. We clarfied in the paper.

---

### Official Review · AnonReviewer4 · 2020-11-08
**Review4**

**Rating:** 6
**Confidence:** 3

**Review:**

This paper studies the lottery ticket hypothesis, which says that there is an underlying sub-network (lottery ticket) in a neural network such that if we train it, we will obtain a better test accuracy compared to the original network. The authors develop a theoretical validation of the improved generalization error of the lottery ticket. Similar to many theoretical results for neural networks, several assumptions have been made. For example, it is assumed that the underlying function, which we try to learn, can be entirely captured by a sparse neural network. This assumption indicates that for some underlying sub-graph, we can learn with zero generalization error. They provide empirical validation for their results as well.

The authors consider a significant problem. Pruning techniques allow us to enjoy high accuracy while reducing neural networks' memory and computational running time. These techniques have been studied for a long time, but mostly from an empirical perspective. I believe understanding the pruning problem's theoretical aspects will help us develop better algorithms for pruning as well.

The results are clearly explained, and the paper is well-written.

Comments:

In empirical cases, the pruned network's initialization is often set to the weights obtained in the training of the more extensive network. Is this technique used in your comparisons in the numerical evaluation?

It is assumed that the covariates x_i's are coming from a Gaussian distribution, implying that the label y_i is a mixture of truncated Gaussians. (before adding noise). While this assumption has been made in previous works, the role of this assumption is not well-explained.

I am not sure whether the underlying assumption trivialized the problem. It is assumed that the function f, which we try to learn, can be captured by a sparse one-hidden-layer neural network. This sparse sub-network would essentially be the lottery-ticket of the fully connected network. Then it is claimed that if we know the structure of this sparse sub-network (i.e., known mask matrix M), then it means we are learning in a lower dimension of parameters compared to the case of a complete network. Thus, it is not surprising to see that the SGD algorithm converges faster. In other words, by knowing M, we already set a large portion of parameters to their optimal values. Thus, it is easier to find the optimal solution to the minimization problem.


The ultimate goal of the pruning problem is to get the sub-structure and train it efficiently. Does your result affect how one could potentially improve the current pruning techniques (such as IMP)? Or can we say anything about the algorithm's performance when a large fraction of the sub-network is picked correctly?

---

> ### Author Response · Authors · 2020-11-21
> **Summary of the changes for the paper, and the replies to the first two comments.**
>
> Thank you for the comments and suggestions. Here are some major revisions in the papers:
>
>
> $\bullet$ We have revised the problem formulation to clarify our setup.
> Our setup can be interpreted by a "teacher-student" mode,
> where the training data are generated from a teacher network, and we learn on a student network with the same number of neurons as the teacher network. We do  not require the student network to have the same architecture as the teacher network. As long as the teacher network can be viewed as a sub-network of the student network by pruning weights, our method finds the ground-truth model with a sufficient number of samples.  We study how the architecture of the student network affects the learning rate and sample complexity in this paper.
>
> $\bullet$ Following the reviewers comments, we added one section (Section 4.1) to discuss the guarantee of obtaining a proper student network architecture from a fully connected dense network by magnitude pruning. The main idea is that after some iterations of learning on the dense network using Algorithm 1, the weights that correspond to a non-existing edge in the teacher network becomes sufficient small (in magnitude), then magnitude pruning removes edges correctly.
>
> The pointwise responses to the comments are summarized in the following:
>
> $\bullet$ Q1: In empirical cases, the pruned network's initialization is often set to the weights obtained in the training of the more extensive network. Is this technique used in your comparisons in the numerical evaluation?
>
> $\bullet$ A1: In our numerical evaluation, we set the pruned network's initialization in the following two methods. One way is that the initial point is obtained by truncating the iterate trained in a more extensive network as you said. The other way is to set it as a random point near the ground truth.
>  To be specific,
>  when conducting the experiments as shown in Figures 6 and 7, the initialization is set by truncating the smallest entries of the weight matrix that obtained in training the more extensive network, and the numerical results suggest that our algorithm works properly.
>
> For Figures 3-5 we directly initialize   randomly in a region around the ground truth. This reduces the computational time. For example, to compute the sample complexity in Figure 4,   we need to test many pairs of $n$ and $d$. The convergence rate in Figure 2  and the estimated model error in Figure 4 do not depend on the exact initiation as long as it is in the local convex region.
>
> $\bullet$ Q2: It is assumed that the covariates $x_i$'s are coming from a Gaussian distribution, implying that the label $y_i$ is a mixture of truncated Gaussians. (before adding noise). While this assumption has been made in previous works, the role of this assumption is not well-explained.
>
> $\bullet$ A2: Thanks for your comments, and we apologize that the rules of the assumption were missed in the original version.
> Overall,  the assumption is critical in most part of the theoretical analysis.  A major part of the proof is to bound the population risk function, which is an expectation over the distribution of input data. Hence, to guarantee the population risk function analyzable, the distribution of input cannot be arbitrary. In addition, Gaussian distribution guarantee a nice landscape of the population risk function, i.e., the local convexity region is large enough.
> Another part is for tensor initialization. The major idea for tensor initialization is to utilize the high-order momentum to characterize the angle and the magnitude of the weights vector $w_j$'s.
>  Gaussian distribution or at least rotation invariant distribution is necessary to separate the information of the angle and the magnitude, which simplifies the calculation. The analysis might be extended to other distributions such as non-standard Gaussian or Gaussian mixture models, and we will leave that for future work.
>
> In the paper, we have added some intuitive discussion for the necessity of Gaussian assumption in the footnote 3 of page 4.

---

> > ### Author Response · Authors · 2020-11-21
> > **Replies to the last two comments**
> >
> > $\bullet$ Q3: Whether the underlying assumption trivialized the problem. It is assumed that the function f, which we try to learn, can be captured by a sparse one-hidden-layer neural network. Thus, it is not surprising to see that the SGD algorithm converges faster. In other words, by knowing M, we already set a large portion of parameters to their optimal values. Thus, it is easier to find the optimal solution to the minimization problem.
> >
> > $\bullet$ A3: Thanks for the comment. We realize from comment that our original submission was not written clearly and might lead to confusion. We revised our setup and added a subsection 4.1 to address your question.
> >
> > First, we need to clarify that we consider a teacher-student setup where the data are generated from a teacher network with $r^*$ weights per neuron. We learn on a student network with $r$ weights per neuron. We only assume $r^* \leq r \leq d$ throughout the paper, and $r^*$ and $r$ can take the entire spectrum of weight density, from sparse to dense network. Thus, our formulation is associated with a quite general case.
> > We removed the statement about sparsity that could lead to confusion.
> >
> > Second, the student network does not need to be the same as the teacher network. As long as the teacher network is a subnetwork of the student network, our results in Theorems 1 and 2  applies. In other words, we do not assume knowing the mask of the teacher network exactly.
> >
> > Third, we added a subsection 4.1 to discuss how to obtain a proper student network from a fully connected neural network using our Algorithm 1 and magnitude pruning. We can train on a fully connected neural network using tensor initialization and iterate with $T_1$ iterations, then apply magnitude pruning to prune the network. We show that the teacher network is indeed a subnetwork of the resulting pruned network. Then this pruned network can be used as the student network discussed in this paper. Please see section 4.1 in the paper for details.
> >
> > Lastly, we would like to emphasize that although it might be intuitive that training on a pruned network is faster than training on a fully-connected network, to the best of our knowledge, there is no theoretical characterization of this intuition explicitly. Our work provides an explicit characterization of the impact of pruning on the sample complexity and the learning rate. The sample complexity decreases, and the learning becomes faster when $r$ in the student network decreases. If the teacher network has a small $r^*$, one can find a proper student network (using our approach in Section 4.1) with a small $r$ such that the learning performance on the student network is much improved over training on a fully connected network.
> >
> > $\bullet$ Q4: The ultimate goal of the pruning problem is to get the sub-structure and train it efficiently. Does your result affect how one could potentially improve the current pruning techniques (such as IMP)? Or can we say anything about the algorithm's performance when a large fraction of the sub-network is picked correctly?
> >
> > $\bullet$ A4: Thanks for the comment. Regarding how to get the right sub-structure, we added a subsection 4.1 to show that combining our Algorithm 1 (with a few iterations only) and magnitude pruning, we can obtain a desirable sub-network from a fully connected network. Then one can run Algorithm 1 again on the sub-network to train the model efficiently.
> >
> > Regarding the dependence of the performance of our algorithm on the sub-network structure,  we clarified in section 2 that we consider a teacher-student setup. The teacher network generates the output from the input, and each neuron connects to at most $r^*$ input features, and $\Omega_j^*$ denotes the set of indices of the input features to which the $j$-th neuron is connected. We train on a  student network. Each neuron connects to at most $r$ input features, and $\Omega_j$ denotes the set of indices of the input features to which the $j$-th neuron is connected. Then, as long as $r\ge r^*$ and $\Omega_j \supseteq \Omega^*_j$, which means that  the teacher network is a sub-network of the student network, the our theorems hold. Note that we do not require the student network to have the same architecture as the teacher network.

---

> > > ### Comment · AnonReviewer4 · 2020-11-25
> > > **Response to your replies**
> > >
> > > I want to thank the reviewer for their detailed response. I have read your response and the changes you had made in your paper.
> > >
> > > As I mentioned earlier, if you know the optimal sub-network structure, the results may not be surprising. In section 4.1, you wrote one could find a proper student network. However, I have some concerns about this part (similar to the ones raised by "AnonReviewer1").  I might be missing something, but the number of samples you spend in the first step to find a proper sub-network, \tilde{O}(d* poly(k) * log 1/eps) is basically enough to run your algorithm on the larger network without no pruning. That is, by the time you find the student network, you have already trained the larger network. Thus, combining these two steps does not seem to be helpful.

---

> > > > ### Author Response · Authors · 2020-11-25
> > > > **Further response on the proper student network architecture**
> > > >
> > > > Thank you very much for your reply.
> > > >
> > > > First, the sample complexity bound of $\mathcal{O}(d \log 1/\varepsilon)$ in the two-stage approach in Section 4.1 is not tight due to the limitation of the current proof techniques for nonconvex optimization. We believe the actual sample complexity is much smaller, and we are still working on this direction to see if the bound can be improved.
> > > >
> > > > Second,  compared with training directly on a dense network, using the two-stage approach to first find the pruned network structure requires a less number of iterations in total. That is because Theorem 2 implies that the second stage converges much faster, and a simple calculation will show the overall number of iterations required is reduced by the two-stage approach compared with training on the dense network. In fact, a formal characterization of the impact of $r$ on the convergence rate (as shown in Theorem 2) is one of the major contributions of this paper.
> > > >
> > > > Third,  we want to clarify that the main results in Theorems 1 and 2 only require the student network to cover the teacher network rather than knowing the teacher network exactly. We understand your concern that this assumption might be strong, nevertheless, the original lottery ticket hypothesis (LTH) and the recent work on linear model connectivity (Frankle et al. 2019b) suggested that there might exist an optimal sub-network with improved generalization over the dense network. This assumption is also needed for tractable analysis. That is, to quantify the improvement analytically,  we need to make sure that the student network has a higher (or the same) expressive power than the teacher network (namely, in LTH, the solution to an optimal sparse network exists), and thus there indeed exists a weight matrix with the student model that can approximate the mapping from input to output.  Many studies have suggested that iterative magnitude pruning is a practical way to find the "winning" mask that satisfies LTH. We agree that providing theoretical insights on how to find a proper sub-network is critical in pruning, however, exploring the reason why training the pruned network is better than training the dense network is also important. The latter is the major focus of this paper.
> > > >
> > > > Jonathan Frankle, Gintare Karolina Dziugaite, Daniel M Roy, and Michael Carbin. Linear mode connectivity and the lottery ticket hypothesis.arXiv preprint arXiv:1912.05671, 2019b. (https://arxiv.org/pdf/1912.05671.pdf)

---

### Author Response · Authors · 2020-11-23
**Looking forward to reviewers' reply**

Dear reviewers,

As Discussion Stage 2 is about to end, we are looking forward to your reply. Please kindly let us know if our response has addressed your initial questions. We appreciate your input and are happy to discuss any follow-up questions. Thank you!

---

### Decision · Program_Chairs · 2021-01-07
**Final Decision**

**Decision:**

Reject

**Comment:**

Even though the authors revised the problem formulation, the paper seems not ready for publication. The assumptions are still too strong (The learning algorithm assumes knowledge of the sparsity mask). The proof technique also heavily relies on Zhong et al'17 without properly highlighting the difference.